# Antibiotic Resistance, Virulence Factors, Phenotyping, and Genotyping of *E. coli* Isolated from the Feces of Healthy Subjects

**DOI:** 10.3390/microorganisms7080251

**Published:** 2019-08-10

**Authors:** Stefano Raimondi, Lucia Righini, Francesco Candeliere, Eliana Musmeci, Francesca Bonvicini, Giovanna Gentilomi, Marjanca Starčič Erjavec, Alberto Amaretti, Maddalena Rossi

**Affiliations:** 1Department of Life Sciences, University of Modena and Reggio Emilia, via Campi 103, 41125, Modena, Italy; 2Department of Pharmacy and Biotechnology, Alma Mater Studiorum-University of Bologna, Via Massarenti 9, 40138 Bologna, Italy; 3Unit of Microbiology, Alma Mater Studiorum-University of Bologna, S. Orsola-Malpighi Hospital, Via Massarenti 9, 40138 Bologna, Italy; 4Department of Biology, Biotechnical Faculty, University of Ljubljana, Jamnikarjeva 101, 1000 Ljubljana, Slovenia

**Keywords:** *Escherichia coli*, typing, gut microbiota, PFGE (pulsed-field gel electrophoresis), virulence, antibiotic resistance, conjugation, curli, co-occurrence

## Abstract

*Escherichia coli* may innocuously colonize the intestine of healthy subjects or may instigate infections in the gut or in other districts. This study investigated intestinal *E. coli* isolated from 20 healthy adults. Fifty-one strains were genotyped by molecular fingerprinting and analyzed for genetic and phenotypic traits, encompassing the profile of antibiotic resistance, biofilm production, the presence of surface structures (such as curli and cellulose), and their performance as recipients in conjugation experiments. A phylogroup classification and analysis of 34 virulence determinants, together with genes associated to the *pks* island (polyketide-peptide genotoxin colibactin) and conjugative elements, was performed. Most of the strains belonged to the phylogroups B1 and B2. The different phylogroups were separated in a principal coordinate space, considering both genetic and functional features, but not considering pulsed-field gel electrophoresis. Within the B2 and F strains, 12 shared the pattern of virulence genes with potential uropathogens. Forty-nine strains were sensitive to all the tested antibiotics. Strains similar to the potential pathogens innocuously inhabited the gut of healthy subjects. However, they may potentially act as etiologic agents of extra-intestinal infections and are susceptible to a wide range of antibiotics. Nevertheless, there is still the possibility to control infections with antibiotic therapy.

## 1. Introduction

*E. coli* are permanent colonizers of the human microbiota that may persist as gut commensals without inducing any intestinal or extraintestinal infections. On the other hand, certain *E. coli* strains may instigate infections not only in the gut but also in other districts, such as those caused by the extraintestinal pathogenic *E. coli* (ExPEC) [1]. The commensalism or virulence of *E. coli* derives from a complex balance between the whole status of the host and the presence and expression of virulence determinants. Commonly, ExPEC strains reside as harmless commensals in the gut of healthy subjects. However, these strains can cause an infection in compromised patients, in case they reach a usually sterile extraintestinal site, such as the urinary tract [2]. The gut microbiota, therefore, act as a powerful reservoir of ExPEC strains potentially responsible for infections, with pathogenic and commensal *E. coli* generally differing in terms of their phylogenetic backgrounds and virulence attributes [3].

The strains that innocuously colonize the intestine of healthy subjects may differ from those that are prone to cause diseases, especially those in possession of accessory traits that confer fitness and competitiveness and shape a specific relationship with the host. The virulence capability of *E. coli* depends on adhesion, biofilm formation, attachment, acquirement of nutrients, competition with other bacteria, toxin production, and avoidance or subversion of host defense mechanisms [4].

Adherence to the epithelial cells is mediated by surface structures or molecules, like fimbrial and afimbrial adhesins, curli, and outer membrane proteins encoded by the *pap* cluster and other genes (*afa*/*draBC*, *fimH*, *focG*, *gafD*, *hra*, *iha*, *sfa*/*focDE*, *sfaS*, and *yfcV*) [5]. Furthermore, *E. coli* exploits several mechanisms of iron uptake that are associated with siderophores and other binding proteins encoded by *chuA*, *fyuA*, *ire*, *iroN*, and *iutA* [5,6]. Indeed, like other pathogens, needing iron for metabolism, *E. coli* must face the host’s response to infection, which involves a reduction in the amount of iron available via a decrease of intestinal iron absorption, the synthesis of additional iron proteins, and shifting iron from the plasma pool into intracellular storage. *E. coli* virulence is also enhanced by the production of toxins (e.g., cytotoxic necrotizing factor 1, autotransporter toxins, and alpha hemolysin) that target the cell’s skeleton, metabolism, or cytoplasmic membrane [5,7].

Genetic exchange increases the success of commensals in invasion, intracellular survival, and spread, providing them with increased fitness and versatility. Hence, the boundary between commensals and pathogens is made fainter by horizontal gene transfer. Mobile genetic elements, such as transposons, plasmids, and insertion sequences, contribute to the plasticity of the *E. coli* genome, resulting in an extremely large pangenome of more than 16,000 genes [8]. Moreover, horizontal gene transfer favors the diffusion of antimicrobial resistance (AR) among both *E. coli* and other commensals, thereby enlarging the spectrum of resistance and promoting epidemiological success, with a bloom in worldwide public health concern associated with the misuse of antibiotics. In particular, conjugation is one of the most important ways for genes to exchange in prokaryotes, leading to genetic variation within a species and the acquisition of new traits. This process requires complex circuits that regulate the transcription of conjugation genes, the assembly of conjugative pili, the formation of the mating pore connecting donor and recipient cells, and the enzymatic processing of plasmid DNA to be transferred [9].

Research interest has been manly focused on characterizing virulent clinical *E. coli.* [10], whereas strains isolated from healthy subjects have been mainly investigated in comparative studies with patients affected by specific diseases [11,12]. A few studies specifically describing the intestinal *E. coli* of healthy subjects mainly focused on antibiotic resistance, without performing a thorough genetic and phenotypic analysis of the strains [13,14,15]. The present study aimed to deeply characterize the population of *E. coli* isolated from the feces of 20 healthy adults in order to determine whether the relationship between PFGE genotyping, phylogroups, genetic determinants, and functional features can be established. A set of 51 strains was analyzed and characterized according to a generally accepted phylotype classification of *E. coli* that includes seven phylogroups of *E. coli sensu stricto* (A, B1, B2, C, D, E, and F) [16,17]. Thirty-four virulence determinants were searched, together with the *pks* island claimed as a concurrent cause in the development of human colorectal cancer [18], the gene *traD*, as an indicator of the presence of conjugative elements [19], the genes encoding E7 colicin (*colE7*), and the immunity protein (*immE7*) [20]. Phenotypical characterization of the isolates included the AR profile, the capability to form biofilms, and the presence of the surface structures involved in adherence.

## 2. Results

### 2.1. Quantitation, Phyloptyping, and Genotyping

*E. coli* were quantified in the feces of 20 healthy subjects by enumeration onto selective plates and through qPCR. These techniques yielded consistent values (*p* = 0.34), ranging from 3.4 × 10^5^ to 9.3 × 10^7^ cfu, or cells per g of feces (Figure 1; Appendix A). The ratio between the *E. coli* and total bacteria ranged from 5.6 × 10^−6^ to 1.9 × 10^−3^, with a mean of 2.8 × 10^−5^.

Ninety-six colonies per sample were randomly selected and clustered based on their ERIC-PCR and RAPD-PCR profiles (Table 1). When profiles were similar across different hosts, an isolate was picked from each of the subjects. A collection of 51 *E. coli* strains was obtained, also encompassing 10 β-glucuronidase-negative clones isolated from the same samples during a parallel study that aimed to characterize Enterobacteriaceae other than *E. coli* (which were ascribed to this species by a MALDI Biotyper matrix-assisted laser desorption/ionisation time-of-flight (MALDI-TOF) system (Bruker Daltonik GmbH, Bremen, Germany) [21]). Within each sample, one to five biotypes were recognized, with most of the samples (14 out of 20) characterized by a dominant biotype (>80%). According to the XbaI PFGE analysis, the 51 strains were ascribed to 44 different pulsotypes (Figure 2).

The strains were assigned to a phylogroup according to Clermont et al. [17] (Table 1). The most prevalent phylogroup was B2 (14 strains), followed by B1, F, A, D, E, and C (12, 7, 7, 5, 4, and 2 strains, respectively). The strains of phylogroup B2 presented the highest absolute charge and were the most abundant within the *E. coli* population, accounting for the all colonies recovered from several samples. B1 phylogroups were detected at lower concentrations and represented a minor subpopulation (Table 1).

### 2.2. Virulence Factors

Thirty-four genetic determinants encoding virulence factors that potentially enhance the risk of infections were screened by PCR. Most of the strains were positive for *fimH*, *fyuA*, *ompT*, *traT*, *chuA*, and *kpsMTII* (47, 36, 35, 35, 32, and 27 strains, respectively). On the other hand, *afa*/draBC, *cnf1*, *focG*, *hlyD*, *ibeA*, *rfc*, *sfa*/*focDE*, and *sfaS* recurred rarely (≤ 5 strains), while *cdtB*, *gafD*, *pic*, and *vat* were absent in all the strains (Figure 3). Based on virulence factors, all the strains belonging to the phylotypes B2, D (with an exception), E, and F clustered together (Figure 4). Strains ascribed to the phylotypes C were grouped in another cluster that also included the majority of B1 and most of the A isolates. A single B1 strain lay outside both clusters.

### 2.3. Other Genetic Determinants Affecting Fitness and Pathogenicity

E. coli strains were screened for the presence of the *pks* pathogenicity island (Figure 3). The genes *clbB* and *clbN*, utilized as markers, were found in 11 out of 51 strains, 3 ascribed to phylotype A and 8 to B2.

The presence of *traD* was investigated to establish the potential in conjugative DNA exchange. Most isolates were positive to *traD* (32/51), including the the majority of pks-positive strains (Figure 3). *traD* was distributed differently among the phylogroups: most B2, C, D, and F strains harbored the gene, while only a minority within A and B1 did the same. When the same subject harbored more biotypes, some isolates bore *traD* while others did not.

The genes *colE7* and *immE7*, respectively encoding the bactericidal nuclease Colicin E7 and the corresponding immunity protein, were searched. Two strains harbored both the genes.

### 2.4. Antibiotic Susceptibility

Phenotypic antibiotic susceptibility tests were carried out on the 51 *E. coli* strains (Appendix A). Nearly all the isolates were extensively sensitive to the whole set of tested antibiotics (49/51). Only 2 strains recovered from the same fecal sample, belonging to the phylogroups B2 (17.18) and A (17.10), were resistant to gentamycin. *E. coli* 17.10 was also resistant to ciprofloxacin.

### 2.5. Conjugation

The 10 β-glucuronidase-negative *E. coli* strains, belonging to phylotypes B1 (6) and F (4), were challenged as conjugation recipients for receiving the pOX38: Cm plasmid from *E. coli* N4i (Table 1). The plasmid transfer to most of the strains succeeded with high efficiency: all the B1 strains and 2 out of the the 4 F strains acquired the plasmid.

### 2.6. Production of Curli and Cellulose and Biofilm Formation

The production of curli was assessed by observing the colonies grown in CR-containing LB agar plates. B2 and F strains presented the lowest ability to bind CR dye (2/14 and 1/7, respectively), whereas C, D, and E strains seemed to be inclined to produce curli (Table 1). The occurrence of cellulose-like extracellular components was analyzed by growing the isolates on LB plates supplemented with CF. The ability of the strains to bind CF was less frequent (9/51) than their ability to bind CR.

*E. coli* strains were assayed for biofilm formation in LBWS and M9glu. Most of the strains did not form a biofilm (Figure 5). Formation occurred more frequently in LBWS than in M9glu (13/51 and 5/51 strains, respectively). Only two strains, both ascribed to phylotype A (03.73 and 08.10), produced a biofilm in LBWS and M9glu. For five other strains, the score of the biofilm formation ranged between 0.6 and 0.8. All but one of these strains presented a very high standard deviation (>0.5; Figure 5), suggesting that biofilm formation was sensitive to environmental conditions that were hardly controlled by the operator.

### 2.7. Co-Occurrence and Principal Coordinate Analysis (PCoA) Analysis of Genetic and Functional Features

A co-occurrence analysis was applied to the whole set of genetic and function observations (Figure 6). The presence of pks islands and several virulence determinants was significantly associated (*p* < 0.05 or < 0.01) to certain genotypic classifications (such as the ERIC profile, the pulsotype, and the phylogroup). In particular, the genes encoding some adhesins (*iha*, *papAH*, *papC*, *papF*, and *sat*), iron binding proteins (*chuA*, *fyuA*, and *iutA*), protectins (*kpsMTII* and *traT*), and *malX*, *ompT*, *usp*, and *traD* were significantly associated to phylogroups and presented a significantly positive tendency to co-occur within the same strains (*p* < 0.05 or 0.01).

With regards to the phenotypic features involved in bacterial adhesion, the production curli were significantly associated with the phylogroup (*p* < 0.01). Although biofilm formation was not frequent, curli presented a significant co-occurrence with the biofilm in the LBWS medium.

A Principal Coordinate Analysis (PCoA) was applied to assist in the identification of clusters, taking into account both their genetic and functional features (Figure 7). The first two coordinates were the most informative, accounting for approximately 28.9% and 10.2% of the total observed variances. In the PCoA plot, PCo1 separated B2 and F strains from B1, C, and E ones (negative and positive PCo1, respectively). F strains were grouped in two clusters. With respect to PCo2, most of the A, C, D, and E strains lie in the positive quadrants.

## 3. Discussion

Fifty-one *E. coli* strains were isolated from the feces of 20 healthy adults and genotypically and phenotypically characterized. Most of the strains belonged to phylogroups B1 (12) and B2 (14). Several B2 strains presented a greater attitude to colonization, with higher charges and an overall dominance in the strains belonging to other phylotypes. This result is consistent with the fact that B2 strains showed an enhanced ability to persist in the intestinal microbiota, as demonstrated in infants [22,23]. B1 strains, on the contrary, showed lower performance in colonization, generally representing minor *E. coli* subpopulations within each sample.

Phylogroup classification and PFGE pulsotypes were consistent, but the strains belonging to the same phylogroups were spread over different clades of the PFGE dendrogram, with only D isolates clustered together. PFGE analysis targeted the whole genome, most of which is composed of accessory genes that can be acquired by horizontal gene transfer. The PFGE fingerprint of *E. coli* results from the digestion of a genome, the size of which, including plasmids and prophages, ranges approximately from 4.6 to 5.9 Mbp, encoding from 4200 to 5500 genes [3]. This huge difference is mainly due to the genes associated with bacteriophage elements and involved in virulence or resistance to antimicrobials. The phylotypes recognized in this study were shuffled in PFGE, likely because of the genome evolution, which was mainly due to the recombination and horizontal transfer that broke the old phylotype relationships, thereby emphasizing more recent changes. On the other hand, MLST typing, the gold-standard approach for *E. coli* classification, allows a phylotyping-consistent analysis, because MLST is based on the conserved nature of the housekeeping genes of the core genome [24]. However, when both the genetic and functional features of each strain were elaborated together in this study (Figure 7), the different phylogroups clustered separately in a PCoA space.

The majority of the strains were sensitive to all the tested antibiotics, including amoxicillin plus clavulanic acid and cephalosporins, the most diffused antibiotics for the first management of infections. Two sole exceptions, resistant to gentamycin and ciprofloxacin, were isolated from the same subject. The absence of the AR phenotype does not exclude the presence of AR genes, which may be expressed in vivo or can be involved in the diffusion and spread of AR genes [25]. However, the rapid emergence of resistant bacteria occurring worldwide attributed to the overuse and misuse of antibiotics, healthy subjects harbor an endogenous *E. coli* population still sensitive to a wide range of antibiotics, contrary to the pathogenic strains isolated from different clinical specimens [26,27,28]. The scarce relevance of commensal *E. coli* as an AR carrier disagrees with other studies [29,30,31,32] but is consistent with a recent analysis of metagenomes that investigated AR genes in the gut microbiota of healthy people. As a whole, the results show that commensal *E. coli* may not pose a threat by itself in terms of AR, though major differences are registered among countries [33,34].

The great variability of pathogenicity-associated features suggests that the genetic determinants of virulence had a role in shaping the genome of the isolates. A significant co-occurrence was observed in a set of B2 and F strains for *pap* genes (*papAH*, *papC*, *papEF*), encoding proteins of the fimbria favoring the ascension of the urinary tract and promoting colonization and infection, *sat*, which is associated with a cytopathic secreted autotransported toxin exerting effect, *kpsMTII*, the marker of K2 capsular polysaccharides, which plays a main role in pathogenesis, *chuA* (heme receptor), *yfcV* (Yfc fimbria), along with *fyuA*, *malX*, *usp*, *ompT* (Figure 3; Figure 6). Among them, twelve strains, belonging to the phylogroups B2 (03.25, 04.06, 06.16, 10.17, 12.04, 18.09, and 19.02) and F (08.57, 08.109, 13.20, 16.04, and 16.27), presented a pattern of virulence genes similar to that of the strains responsible for urinary tract infections, on the basis of the presence of the genes *chuA* (heme receptor), *yfcV* (Yfc fimbria), along with *fyuA* (*Yersinia* siderophore receptor) and *ompT* (aspartyl protease) [35]. This reinforces the idea that it is possible that pathogenic *E. coli* potentially inhabit the gut of healthy subjects without instigating infections, although they can act as etiologic agents of extra-intestinal infections, without a clear distinction between commensal *E. coli* and pathogens.

Capsular polysaccharides represent a class of macromolecules contributing to the surface properties. They are involved in important biological processes including adhesion and resistance to the host’s immune responses, such as complement-mediated killing and phagocytosis [5]. Many strains (27/51) carried *kpsMTII*, among which six strains also carried *kpsMTIII*, encoding another capsular antigen. Macromolecules on the surface of bacteria confer ultrastructural stability are important for recognition by, and interaction with, the environment and in pathogenic bacteria form a defensive barrier against the host’s immune system. All but one strain was negative to the *afa*/*draBC* gene, excluding the presence of afimbrial adhesins encoded by the *afa* operons [36]. Likewise, for capsular polysaccharides, curli are involved in adhesion and strengthen *E. coli* defenses, particularly against complement-mediated killing by the host, thereby contributing to the pathogenic potential of strains [37]. The positive relationship between curli and biofilm formation was confirmed in the present study, although the latter feature was not frequent among the observed strains. Interestingly, the phylogroups B2 and F, which include the strains with the largest set of potential pathogenic determinants, are less prone to produce curli, a condition that may increase their susceptibility to host defenses.

*E. coli* pathogenicity islands serve as integration sites for genetic elements. Thus, virulence genes and AR determinants can be rapidly and simultaneously acquired, changing commensal strains into major risk agents [38]. Nineteen stains, representing most of the B2 or F isolates, harbored a PAI island, which produced a positive result for the maltose- and glucose-specific component IIa of the phosphoenolpyruvate dependent phosphotransferase system (*malX*). Furthermore, all the isolates also carried the gene *usp*, encoding a putative bacteriocin generally located on the PAI [31]. Eleven out of 51 strains, belonging to the phylogroups A and B2, harbored the 54-kb polyketide synthases *pks* pathogenicity island, encoding the multi-enzymatic machinery for the synthesis of a peptidepolyketide hybrid genotoxin called colibactin [39]. *hlyF*, encoding for the toxin α-emolysin [40], occurred in other strains, which were spread over most of the phylotypes.

The potentiality of the isolates to exchange genetic material was assessed by searching the gene *traD* and challenging the β-glucuronidase negative *E. coli* strains as recipients in conjugation experiments. *traD* is a key gene in the conjugation process, encoding a main component in the transferosome of type IV secretion systems [19]. The presence of *traD* has been chosen to screen the presence of the conjugation apparatus and the potentiality of the strains to exchange DNA. The majority of the strains (32/51) carried *traD*, regardless of the phylotype group. When more than a single *E. coli* strain was identified in the same subject (Table 1), some isolates encoded *traD* and others did not. The 9 strains harboring the *iss* gene (responsible for the increased survival of bacteria in the serum and generally carried by the big virulence plasmid, ColV) were all positive to *traD*, according to the fact that ColV plasmids are usually conjugative [41]. The majority of β-glucuronidase-negative strains, assayed as recipients in the conjugation experiment, received the plasmid (8/10), confirming their receptive aptitude for the genetic exchange and DNA shuffling of commensal *E. coli*. Interestingly, 2 of the 8 transconjugant strains harbored *traD*, a putative marker of a conjugative plasmid that is expected to exclude the acquirement of another plasmid through conjugation.

The fitness and competitiveness of the bacteria can be improved by bacteriocin production, which is generally associated with the counterpart immunity protein, thereby protecting the producing host cell from the lethal action of the bacteriocin. Colicins are plasmid encoded toxins produced by *E. coli* under conditions of stress and able to kill related bacteria competing for niches and nutrients. Colicin E7, and its immunity counterpart, ImmE7, have been used to develop bacterial conjugation-based antimicrobial agents [42,43]. The antibacterial activity of the colicin ColE7, bacterial “kill”–“anti-kill” antimicrobial system has been determined, thereby offering new perspectives in the development of *E. coli* targeted antimicrobials. The strains isolated in this study were screened for the presence of the *colE7* and *immE7* genes, in order to investigate the susceptibility of commensal strains to the action of recombinant antimicrobials. Most of the strains were negative for both the genes, suggesting that, if necessary, they could be potentially vulnerable to this new approach.

## 4. Materials and Methods

### 4.1. Isolation and Enumeration and of E. coli

Fresh fecal samples were collected from 20 healthy adult subjects, who were all caucasians: 10 males and 10 females aged 35 to 45, following a western omnivore diet, who had not been treated with prebiotics and/or probiotics for 1 month and antibiotics for 3 months. They did not have any relevant diseases or drug treatments in their medical histories. These subjects were enrolled as volunteers among the employees of the University of Modena and Reggio Emilia and their relatives and were not in a relationship with the researchers. All human clinical samples were collected after the subjects gave written informed consent for their participation in the study. All personal data used in the study were anonymized.

Fresh fecal samples were homogenized (10% *w/v*) and serially diluted in isotonic Buffered Peptone Water (Sigma, Steinheim, Germany). Then, they were spread onto HiCrome Coliform Agar (HCCA, Sigma) and incubated. After 24 h at 37 °C, the putative *E. coli* colonies were checked for indole production with Kovac’s reagent (Sigma). Colonies that exhibited consistent reactions were ascribed to the species *E. coli*. Isolates were maintained and propagated in a Luria Bertani (LB) medium aerobically at 37 °C.

A total of 96 *E. coli* clones per sample were subjected to ERIC-PCR [44] and RAPD-PCR [45] and clustered into biotypes with a similarity level of 75% using the Pearson correlation coefficient. For each strain, the concentration was determined by multiplying the relative abundance and total plate count of *E. coli*.

For qPCR quantification, the feces were suspended in PBS pH 7.8 (10% *w/w*). Bacterial gDNA was extracted with the QIAmp DNA Stool Mini Kit (Qiagen, Hilden, Germany), quantified with a NanoPhotometer *p*-Class (Implen GmbH, Munchen, Germany), diluted to 2.5 ng/µL in a TE buffer (pH 8), and subjected to qPCR analysis, with primers targeting eubacteria and *E. coli*. [46]. The mixture contained 10 µL of SsoFast EvaGreen Supermix, 4 µL of each 2 µM primer, and 2 µL of the template. qPCR reactions were carried out with the CFX96 Real-Time System (Bio-Rad Laboratories, Redmond, WA, USA) according to the following program: 98 °C for 2 min; 45 cycles at 98 °C for 0.05 min, 60 °C for 0.05 min, and 95 °C for 1 min; 65 °C for 1 min.

### 4.2. Phyloptyping and PFGE Genotyping

Standard multiplex PCR was used for all the isolates, according to the revisited phylotyping method published by Clermont et al. [17]. PFGE was performed according to the PulseNet protocol [47]. The genomic DNA was digested with 50 U of *Xba*I at 37 °C for 3 h. Macrorestriction fragments were resolved by counter-clamped homogeneous electric field electrophoresis in a CHEF-DRIII apparatus (Bio-Rad, USA). The run time was 24 h at 6.0 V/cm, with initial and final switch times of 2.2 s and 54.2 s, respectively. PFGE images were digitally captured and analyzed with the GelCompare II 6.0 software (Applied Maths NV, Belgium). Dice’s coefficients were calculate based on the band profiles, setting the position tolerance at 1% and the optimization at 1%. A similarity dendrogram was derived using the unweighted pair group method with arithmetic means (UPGMA). Strains were ascribed to the same pulsotype if the PFGE profile possessed >85% similarity.

### 4.3. Virulence Genotyping

All the isolates were screened by multiplex-PCR for a carriage of 34 genes associated to virulence factors: *afa/draBC*, *cdtB*, *chuA*, *cnf1*, *cvaC*, *fimH*, *focG*, *fyuA*, *gafD*, *hlyD*, *hlyF*, *hra*, *ibeA*, *iha*, *ire*, *iroN*, *iss*, *iutA*, *KpsMTII*, *kpsMTIII*, *malX* (*PAI*), *ompT*, *papAH*, *papC*, *papEF*, *pic*, *rfc*, *sat*, *sfa*, *sfaS*, *traT*, *usp*, *vat*, and *yfcV*. Primer sequences, the pool of primers, and amplification reactions were set up according to Johnson et al. [48].

### 4.4. Detection of the Genes traD, colE7, immE7, and of pks Island

A triplex-PCR was utilized to screen the isolates for the presence of the genes *colE7*, *immE7*, and t*raD*, utilizing the primers, colE7-F/colE7-R2 (5′-AAGTCAGATGCTGATGTTGC-3′/5′-ATAGACACCACCATTTTGGC-3′), immE7-F/immE7-R2 (5′-GCAACTGATGATGTGTTAGATG-3′ /5′-TGTTTAAATCCTGGCTTACCG-3′), and traD-F/traD-R2 (5′-GATAAATACATGATCTGGTGCGG-3′/5′-TTGATGGAAAGCAGTGTGTC-3′), respectively, specifically developed in this study. The amplification reaction was carried out in a final volume of 50 µL containing 5 µL of 10X Dream Taq Green Buffer, 5 µL dNTP Mix (2 mM each), 0.25 µL Dream Taq, 9 µL primer mix (2 µM each), and 50 ng of template DNA. The thermocycle was: 95 °C for 5 min, 30 cycles at 95 °C for 30 sec, 58 °C for 30 sec, and 72 °C for 1 min; and 72 °C for 7 min.

The polyketide synthases *pks* pathogenicity island was searched by duplex-PCR, amplifying the *clbB* and *clbN* genes located at the 5′ and 3′ regions of the island, according to Sarshar et al. [49].

### 4.5. Antibiotic Susceptibility

Antimicrobial susceptibility was tested with a Vitek2 semi-automated system (bioMerieux, France). Minimum inhibitory concentrations (MICs) were interpreted according to EUCAST (European Committee on Antimicrobial Susceptibility Testing—www.eucast.org). Clinical breakpoints for the tested antibiotics were expressed as mg/L (S, susceptible; R, resistant): for amikacin, S ≤ 8 and R > 16; for amoxicillin/clavulanic acid, S ≤ 8 and R > 8; for cefotaxime, 1 ≤ 8 and R > 2; for ceftazidime, S ≤ 8 and R > 8; for ciprofloxacin, S ≤ 0.5 and R > 1; for gentamicin, S ≤ 2 and R > 4; for piperacillin + tazobactam, S ≤ 8 and R > 16; and for trimethoprim/sulfamethoxazole, S ≤ 40 and R > 80.

### 4.6. Biofilm and Phenotype Assays

Biofilm formation was assayed by crystal violet (CV) staining, as described in [50]. *E. coli* strains were cultured in LB without NaCl and in an M9 medium (BD Difco, Sparks, Maryland, USA) containing 4 g/L glucose and 0.25 g/L yeast extract (hereinafter referred to as LBWS and M9glu, respectively). The specific biofilm formation (SBF) index was calculated as the ratio between CV absorbance at 570 nm and the culture’s turbidity at 600 nm, setting a threshold of 1. The data reported here are the means of 3 independent experiments, each carried out in triplicate.

Curli and cellulose formation were assayed in LBWS agar plates supplemented with congo red (CR) or calcofluor white (CF), respectively, according to [51]. Red colonies were considered positive to CR. Emission of fluorescence as result of UV exposure (315–400 nm) was considered positive to CF.

### 4.7. Solid Mating Conjugation Experiments

The donor strain *E. coli* N4i pOX38: Cm (N4i: EcN immE7 Gmr; pOX38: Cm: Tra+ RepFIA+ Cmr) was derived from *E. coli* Nissle 1917 [42,43]. The donor was cultured aerobically at 37 °C in LB supplemented with 15 µg/mL gentamycin and 20 µg/mL chloramphenicol, while the recipients were cultured in LB. Overnight donor and recipient cultures were seeded in their respective media (1% v/v) and aerobically incubated for 2 h. One mL of the recipient exponential culture was centrifuged (10,000× *g* for 5 min at 4 °C) and resuspended in 400 µL of the donor exponential culture. The whole volume was spread onto an LB–agar plate and incubated at 37 °C for 4 h, and then the cells were collected from the surface with 1 mL of PBS, serially diluted, and spread onto HCCA to differentiate the recipient and transconjugant from the donor colonies and HCCA supplemented with 20 µg/mL of chloramphenicol to select and differentiate the transconjugant from the donor colonies.

### 4.8. Statistical Analysis

Dice’s coefficient was utilized to gauge the distance between strains, based on the sets of binary data, i.e., the presence/absence of genes or phenotypic properties. Distance matrices between strains were utilized to compute and display UPGMA dendrograms and Principal Coordinates Analysis plots. The co-occurrence of genetic determinants and phenotypic properties were evaluated by means of contingency tables and Cramér’s V metrics, considering the limit of significance as *p* < 0.01 and *p* < 0.05. Statistical analyses were performed with the SPSS Statistics 21 software (Armonk, New York, USA).

## 5. Conclusions

A characterization of *E. coli* isolated from fecal samples of healthy subjects was performed. Several strains had the potential to cause extraintestinal infections because of the presence of genes associated with adhesins, siderophores, and toxins. The recurrence of strains sharing a pattern of virulence genes similar to that of potentially pathogenic strains may pose a health threat. However, a main outcome of this study shows the great sensitivity of the isolates to antibiotics, which are susceptible to the common antimicrobial therapy used for Gram-negative bacteria, such as amoxicillin/clavulanate. This suggests that the *E. coli* populations inhabiting the gut of healthy subjects and of patients are more likely differentiated on the basis of AR than virulence factors, emphasizing the importance of the prudent use of antibiotics in the general population.

## Figures and Tables

**Figure 1 microorganisms-07-00251-f001:**
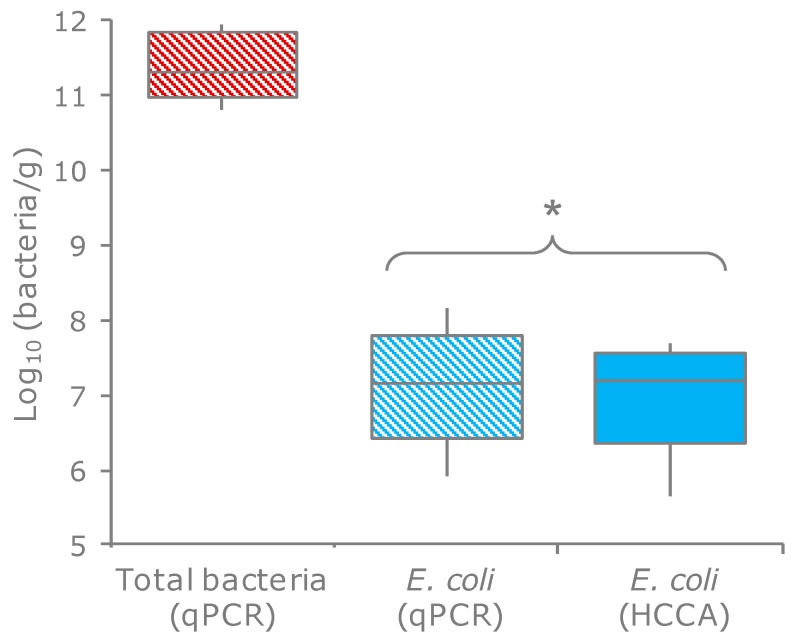
Counts of *E. coli* and total bacteria in the feces of 20 subjects. Boxes indicate the median and 25th and 75th percentiles; whiskers indicate the 10th and 90th percentiles. * indicates the means that did not significantly differ (*p* > 0.05, paired samples t-test). HCCA, HiCrome Coliform Agar (the selective medium for *E. coli*).

**Figure 2 microorganisms-07-00251-f002:**
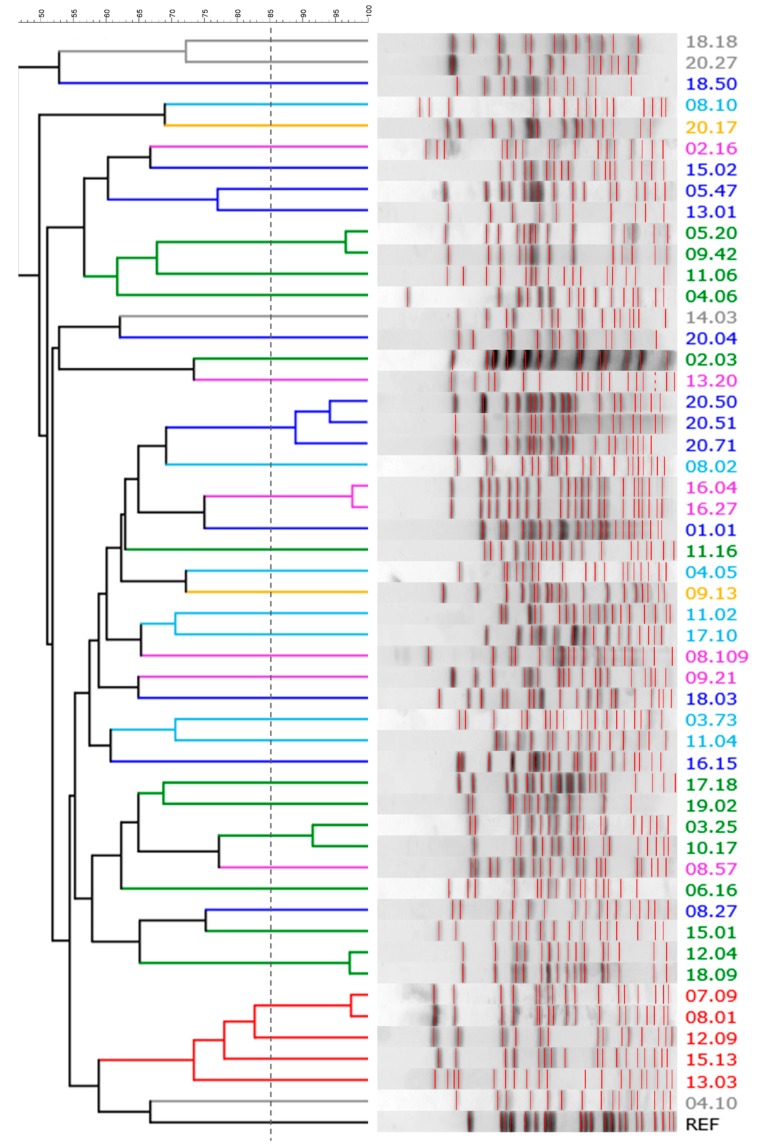
The *Xba*I-PFGE pattern of *E. coli* strains: a UPGMA dendrogram derived from Dice’s coefficients, calculated based on their band profiles. Patterns with similarities >85% (dashed line) were ascribed to the same pulsotype. Strains are colored based on their phylogroup: A, cyan; B1, blue; B2, green; C, yellow; D, red; E, grey; F, pink.

**Figure 3 microorganisms-07-00251-f003:**
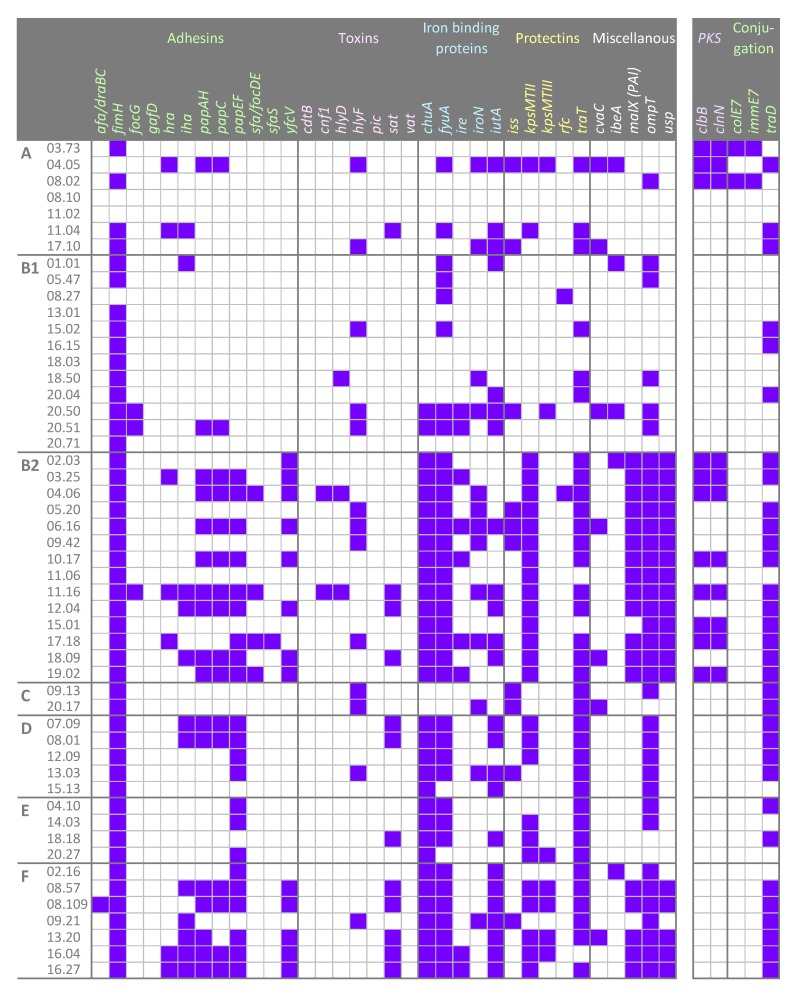
The pattern of genetic determinants in the 51 *E. coli* strains, ordered on the basis of their phylotypes. Virulence genes coding for adhesins, toxins, iron binding proteins, and protectins were screened. The presence of the *pks* pathogenicity island and other genes involved in the conjugation machinery were searched. Purple squares, presence; white squares, absence.

**Figure 4 microorganisms-07-00251-f004:**
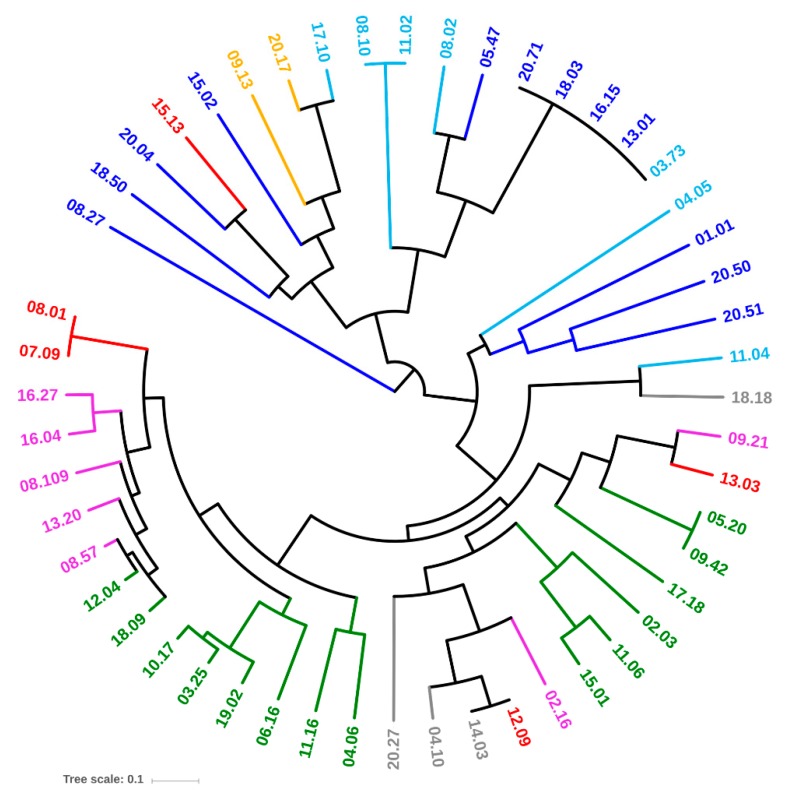
UPGMA dendrogram of *E. coli* strains, computed from Dice’s distance matrix of the virulence determinants. Strains are colored based on their phylogroups: A, cyan; B1, blue; B2, green; C, yellow; D, red; E, grey; F, pink.

**Figure 5 microorganisms-07-00251-f005:**
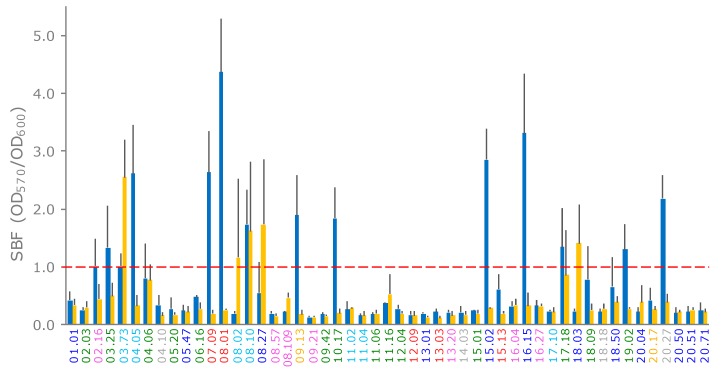
formation of *E. coli* strains on LBWS (blue bars) and M9glu (yellow bars). The specific biofilm formation index (SBF) is calculated as the ratio between the crystal violet absorbance at 570 nm and the culture turbidity at 600 nm, setting a threshold of 1 for biofilm producers (red dashed line). The reported data are means ± standard deviations of at least three independent experiments, each carried out in triplicate. The names of the strains are colored based on their phylogroups: A, cyan; B1, blue; B2, green; C, yellow; D, red; E, grey; and F, pink (hereafter referred to as LBWS and M9glu, respectively).

**Figure 6 microorganisms-07-00251-f006:**
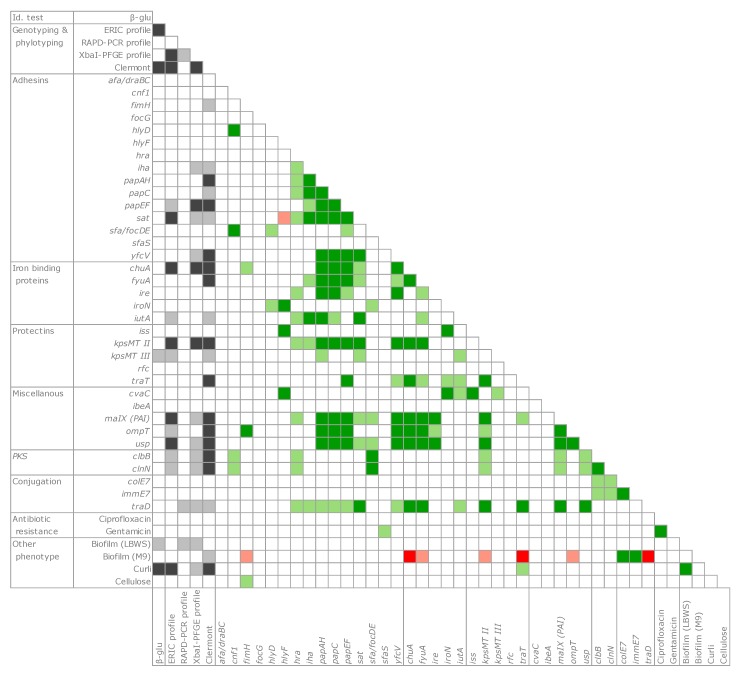
Genetic determinants and phenotypic properties evaluated by Cramér’s V metrics. Association with genotype/phylotype clusters, grey; positive co-occurrence, green; negative co-occurrence, red. Dark and light shades indicate the levels of statistical significance (*p* < 0.01 and *p* < 0.05, respectively).

**Figure 7 microorganisms-07-00251-f007:**
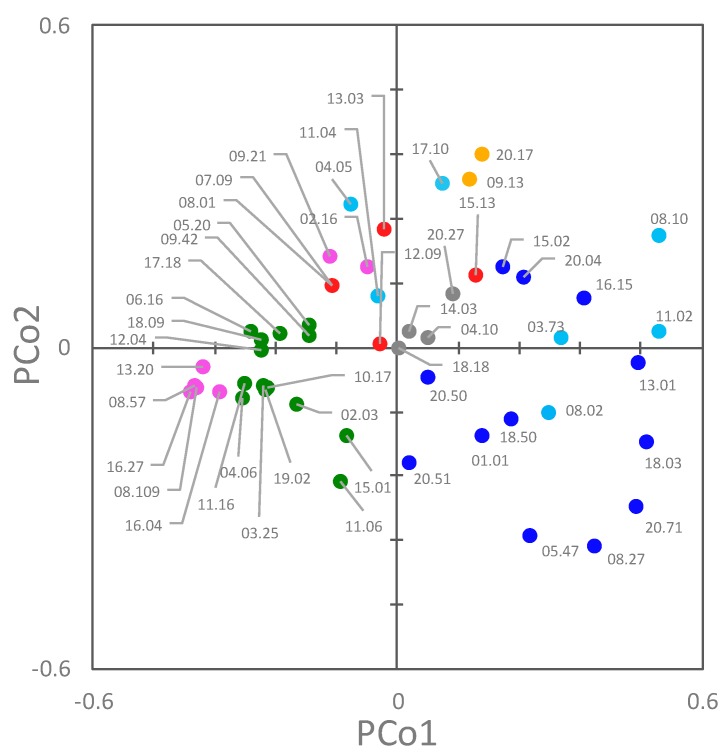
2D Principal Coordinate Analysis (PCoA) visualization of *E. coli* strains, computed from the Dice’s distance matrix of genetic and phenotypic determinants. Strains are colored based on their phylogroup: A, cyan; B1, blue; B2, green; C, yellow; D, red; E, grey; F, pink.

**Table 1 microorganisms-07-00251-t001:** *E. coli* isolates from 20 healthy subjects. The strains were genotyped by ERIC-PCR, RAPD-PCR, and PFGE analysis. The relative abundance represents the ratio of each strain within the set of *E. coli* clones analyzed. The charge of the viable cells has been calculated by multiplying the total *E. coli* count and the relative amount. Phlylotyping was done according to Clermont et al. 2013 [17]. The presence of β-glucuronidase was confirmed by growth of blue colonies on the HCCA medium. The performance of isolates as recipients in conjugation experiments was assessed only in β-glucuronidase negative strains. Biofilm formation was tested on LBWS and M9glu media. Curli and cellulose production was determined.

*E. coli* Strain	Genotyping and Phyloptyping						Phenotype Assays					
	ERIC-PCR			RAPD-PCR Profile	*Xba*I-PFGE Profile	Phylo-Group	β-glu	Conju-Gation	Biofilm		Curli	Cellulose
	Profile	Relative Abundance	Log (cfu/g)						LBWS	M9		
01.01	E01	100%	7.56	R01	P01	B1	+		−	−	−	−
02.03	E02	8%	5.22	R02	P02	B2	+		−	−	−	−
02.16	E03	92%	6.28	R03	P03	F	+		+	−	+	+
03.25	E04	99%	7.52	R04	P04	B2	+		+	−	−	−
03.73	E05	1%	5.53	R03	P05	A	+		+	+	−	−
04.05	E06	5%	4.22	R05	P06	A	+		+	−	+	−
04.06	E07	70%	5.37	R06	P07	B2	+		−	−	−	−
04.10	E08	25%	4.92	R07	P08	E	+		−	−	+	+
05.20	E07	96%	7.78	R02	P09	B2	+		−	−	−	+
05.47	E09	4%	6.40	R08	P10	B1	−	+	−	−	−	−
06.16	E07	100%	7.28	R07	P11	B2	+		−	−	−	−
07.09	E10	100%	6.45	R04	P12	D	+		+	−	+	−
08.01	E11	18%	4.91	R09	P12	D	+		+	−	+	−
08.02	E12	5%	4.37	R10	P13	A	+		−	+	−	−
08.10	E13	5%	4.37	R11	P14	A	+		+	+	+	−
08.27	E14	36%	5.20	R11	P15	B1	−	+	−	+	−	−
08.57	E15	34%	5.18	R12	P16	F	−	+	−	−	−	−
08.109	E16	2%	3.86	R13	P17	F	−	+	−	−	−	−
09.13	E01	92%	7.09	R14	P18	C	+		+	−	+	+
09.21	E17	4%	5.73	R12	P19	F	+		−	−	−	−
09.42	E07	4%	5.73	R12	P09	B2	+		−	−	−	−
10.17	E04	100%	7.62	R12	P04	B2	+		+	−	−	−
11.02	E18	50%	7.67	R15	P20	A	+		−	−	−	−
11.04	E19	9%	6.91	R10	P21	A	+		−	−	−	−
11.06	E07	37%	7.54	R16	P22	B2	+		−	−	−	−
11.16	E20	4%	6.57	R17	P23	B2	+		−	−	−	−
12.04	E13	71%	7.43	R18	P24	B2	+		−	−	+	−
12.09	E11	29%	7.04	R18	P25	D	+		−	−	−	−
13.01	E01	3%	4.98	R19	P26	B1	+		−	−	+	−
13.03	E11	1%	4.47	R11	P27	D	+		−	−	+	+
13.20	E21	97%	6.46	R13	P28	F	−	−	−	−	−	−
14.03	E22	100%	6.85	R13	P29	E	+		−	−	+	−
15.01	E07	9%	6.45	R13	P30	B2	+		−	−	−	+
15.02	E01	87%	7.45	R11	P31	B1	+		+	−	+	+
15.13	E11	4%	6.15	R20	P32	D	+		−	−	+	−
16.04	E21	1%	5.74	R21	P33	F	+		−	−	−	−
16.15	E01	1%	5.74	R22	P34	B1	+		+	−	+	−
16.27	E21	98%	7.73	R23	P33	F	−	−	−	−	−	−
17.10	E23	87%	7.45	R23	P35	A	+		−	−	−	−
17.18	E07	13%	6.63	R24	P36	B2	+		+	−	−	−
18.03	E24	39%	6.25	R25	P37	B1	+		−	+	−	−
18.09	E13	41%	6.28	R25	P24	B2	+		−	−	+	−
18.18	E25	19%	5.93	R13	P38	E	+		−	−	−	+
18.50	E26	1%	4.70	R21	P39	B1	−	+	−	−	−	−
19.02	E07	100%	7.19	R14	P40	B2	+		+	−	−	−
20.04	E27	12%	6.39	R26	P41	B1	+		−	−	+	−
20.17	E28	5%	6.01	R12	P42	C	+		−	−	+	−
20.27	E29	53%	7.03	R27	P43	E	+		+	−	+	+
20.50	E30	22%	6.65	R04	P44	B1	−	+	−	−	−	−
20.51	E31	4%	5.91	R06	P44	B1	−	+	−	−	−	−
20.71	E09	4%	5.91	R06	P44	B1	−	+	−	−	−	−

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
