# Peer review of "Antibiotic Resistance, Virulence Factors, Phenotyping, and Genotyping of E. coli Isolated from the Feces of Healthy Subjects"

_microorganisms, 2019, doi:10.3390/microorganisms7080251_

Round 1

Reviewer 1 Report

In the present study, the authors genotyped and analyzed 51 E. coli strains from feces of healthy humans, e.g. regarding their ability to form biofilms, produce curli & cellular and important virulence factors. 

The ms is well-written and the study was performed well. However, I am not convinced of its impact. 

Major comments:

In Line 64-66, the authors state: " but nowadays it has been extended also towards fecal isolates from healthy subjects and to environmental strains". Do the authors refer to published studies? If so, please reference and explain how the present study contributes novel insights. If this is the very first study of this kind, then it certainly deserves publication with minor revisions. If similar studies of E. coli feces samples have been published, it has to be explained how the present study expands the already existing knowledge. 

Simply the presence of the virulence factors may not mean a lot. The authors could compare the strain with the most virulence factors and the one that harbors the least e.g. in an infection model. If this is not possible, the could test if select genes are expressed and compare the expression. This could help to understand whether and to what extend the virulence factors may contribute to the pathogenicity of the strain.

Minor comments:

1) The abstract should not contain abbreviations

2) Line 47-49: The authors introduce the reader to the different virulence traits that play a role for the likelihood of E. coli to cause a disease. It would be helpful if they could go into more detail and present the most important gene products.

3) Fig. 1: Explain HCAA.

4) Table 1: Biofilm data should be visualized in a bar diagram

5) In general: Figures need to be explained in more detail, both in the Figure legend and in the text.

Author Response

>The authors would like to thank the Reviewers for the constructive comments and criticisms which have effectively contributed to improve the paper. We did our best to properly respond to each point raised by the reviewers. 

REVIEWER1

Comments and Suggestions for Authors

In the present study, the authors genotyped and analyzed 51 E. coli strains from feces of healthy humans, e.g. regarding their ability to form biofilms, produce curli & cellular and important virulence factors. 

The ms is well-written and the study was performed well. However, I am not convinced of its impact. 

Major comments:

In Line 64-66, the authors state: " but nowadays it has been extended also towards fecal isolates from healthy subjects and to environmental strains". Do the authors refer to published studies? If so, please reference and explain how the present study contributes novel insights. If this is the very first study of this kind, then it certainly deserves publication with minor revisions. If similar studies of E. coli feces samples have been published, it has to be explained how the present study expands the already existing knowledge.

>The paragraph has been rewritten in order to better clarify what is already reported in the literature and which is the novelty of this study on healthy subjects. Literature citations has been improved, too.

Simply the presence of the virulence factors may not mean a lot. The authors could compare the strain with the most virulence factors and the one that harbors the least e.g. in an infection model. If this is not possible, the could test if select genes are expressed and compare the expression. This could help to understand whether and to what extend the virulence factors may contribute to the pathogenicity of the strain.

>The authors agree with Reviewer1 that the presence of the virulence factors yields only preliminary evaluation and that an infection model or an expression study would be necessary to be more conclusive on the pathogenic potential of E. coli strains. However, such investigations are beyond the aim of this study, mainly focused on the genetic and phenotypic intraspecific diversity among isolates. The evaluation of pathogenicity in a rodent of some of the E. coli isolates and other Enterobacteriaceae, similarly isolated from healthy subjects, is the aim of a parallel study.

Minor comments:

The abstract should not contain abbreviations

>The abstract has been extensively modified also removing abbreviations.

2) Line 47-49: The authors introduce the reader to the different virulence traits that play a role for the likelihood of E. coli to cause a disease. It would be helpful if they could go into more detail and present the most important gene products.

>Many details have been added about the role of the most relevant virulence genes in E. coli.

3) Fig. 1: Explain HCAA.

>Fig.1 legend has been amended explaining the abbreviation HCCA.

4) Table 1: Biofilm data should be visualized in a bar diagram

>Table.1 legend has been modified and the suggested bar plot was added in order to clearly illustrate biofilm results (new Figure 5).

5) In general: Figures need to be explained in more detail, both in the Figure legend and in the text.

>The manuscript text has been improved according to Reviewer1 suggestions and the figures have been deeply explained modifying the legends.

Reviewer 2 Report

This work intend to study deeply the different microorganisms observed in the healthy population.

However I have a major problem when reading the article why are the results presented in section 2. before the methods ? this is particularly hard to read and should be corrected because it does not make sense.

In The Methods how were the 20 healthy individuals chosen ? Were the volunteers ? we don't know anything about their genetic background (ethnic origin, first degree past medical history). Are we sure they did not suffer from any medical past history ? because some are 45 years old.

In terms of results, about the fact they do not share resistance in the antibiotic susceptibility test and the link with AR genes you stated "49 strains were sensitive to all the tested antibiotics, indicating that AR genes did 28 not heavily shape the genome of commensals from healthy subjects", I would recommend more precaution. Indeed the close link between expression of genes and phenotype might be hard to demonstrate (https://www.ncbi.nlm.nih.gov/pubmed/25517437) Also maybe you would see AR genes of resistance if you were working on strains of Klebsiella, so the point of only stating about E. coli does not make sense for me.

In the same way, you suggest that "Potentially pathogenic strains inhabited the gut of healthy subjects without causing infections" because they were susceptible to antibiotics. Besides you support this idea by citing reference 25-27. However, this is false, indeed you can have very virulent strains that are wildtype. For instance this is the case for Staphylococcus aureus ST 21 which is hypervirulent and susceptible to penicillin : https://www.ncbi.nlm.nih.gov/pubmed/26445995

Conversely, i believe that the conclusion to your work is that despite some virulence factors, "healthy individuals" are still susceptible to common antimicrobial therapy used for gram-negative bacteria such as amoxicillin/clavulanate, which emphasizes the importance of the good use of anitbiotics in the general population.

Author Response

>The authors would like to thank the Reviewers for the constructive comments and criticisms which have effectively contributed to improve the paper. We did our best to properly respond to each point raised by the reviewers. 

REVIEWER2

Comments and Suggestions for Authors

This work intend to study deeply the different microorganisms observed in the healthy population.

However I have a major problem when reading the article why are the results presented in section 2. before the methods ? this is particularly hard to read and should be corrected because it does not make sense.

>The manuscript text has been checked, in section 2 only materials and methods has been reported, results have been collected in section 3.

In The Methods how were the 20 healthy individuals chosen ? Were the volunteers ? we don't know anything about their genetic background (ethnic origin, first degree past medical history). Are we sure they did not suffer from any medical past history ? because some are 45 years old.

>The Materials and Methods has been improved reporting the requested information about volunteers’ enrolment.

In terms of results, about the fact they do not share resistance in the antibiotic susceptibility test and the link with AR genes you stated "49 strains were sensitive to all the tested antibiotics, indicating that AR genes did 28 not heavily shape the genome of commensals from healthy subjects", I would recommend more precaution. Indeed the close link between expression of genes and phenotype might be hard to demonstrate (https://www.ncbi.nlm.nih.gov/pubmed/25517437) Also maybe you would see AR genes of resistance if you were working on strains of Klebsiella, so the point of only stating about E. coli does not make sense for me.

>According to  Reviewer2, the sentence “49 strains were sensitive to all the tested antibiotics, indicating that AR genes did 28 not heavily shape the genome of commensals from healthy subjects" has been substitute with the phrase “The absence of the AR phenotype does not exclude the presence of AR genes, that may be expressed in vivo or can be involved in diffusion and spread of AR genes”

In the same way, you suggest that "Potentially pathogenic strains inhabited the gut of healthy subjects without causing infections" because they were susceptible to antibiotics. Besides you support this idea by citing reference 25-27. However, this is false, indeed you can have very virulent strains that are wildtype. For instance this is the case for Staphylococcus aureus ST 21 which is hypervirulent and susceptible to penicillin : https://www.ncbi.nlm.nih.gov/pubmed/26445995

>The authors agree with the Reviewer2: very virulent strains may be susceptible to common antibiotics; one thing does not exclude the other. In the conclusion was stated that “Potentially pathogenic strains inhabited the gut of healthy subjects without causing infections” that is simply an observation: the volunteers were healthy at sampling time and potentially pathogenic strains have been found in their faeces. The recurrence of strains sharing a pattern of virulence genes similar to that of potentially pathogenic strains may pose a health threat, but the sensitivity to antibiotics is not presented as a cause of the healthy state of the volunteers. Anyway, it has to be considered as a favourable peculiarity of these strains in the case of development of an infection: the absence of antibiotic resistance makes it easier to control a potential infection.

Conversely, i believe that the conclusion to your work is that despite some virulence factors, "healthy individuals" are still susceptible to common antimicrobial therapy used for gram-negative bacteria such as amoxicillin/clavulanate, which emphasizes the importance of the good use of anitbiotics in the general population.

>The authors agree with this final statement and have incorporated it in the text to better outline their conclusions

Round 2

Reviewer 1 Report

All comments were addressed

Line 73: mainly instead of manly.